# Imagine Within Practice: Conservative Rollout Length Adaptation for Model-Based Reinforcement Learning

## Abstract

Model-based reinforcement learning (MBRL) algorithms achieve high sample efficiency by leveraging imagined rollouts from a world model for policy optimization. A crucial hyperparameter in MBRL is the *rollout length*, which represents a trade-off between data quality and efficiency by limiting the imaginary horizon. While longer rollout length offers enhanced efficiency, it introduces more unrealistic data due to compounding error, potentially leading to catastrophic performance deterioration. To prevent significant deviations between imagined rollouts and real transitions, most model-based methods manually tune a fixed rollout length for the entire training process. However, the fixed rollout length is not optimal for all rollouts and does not effectively prevent the generation of unrealistic data. To tackle this problem, we propose a novel method called **C**onservative **R**ollout **L**ength **A**daptation (**CRLA**), which conservatively restricts the agent from selecting actions that are rarely taken in the current state. CRLA truncates the rollout to preserve safety when there is a high probability of selecting infrequently taken actions. We apply our method to DreamerV3 and evaluate it on the Atari 100k benchmark. The results demonstrate that CRLA can effectively balance data quality and efficiency by adjusting rollout length and achieve significant performance gains in most Atari games compared to DreamerV3 in the default setting.

## 1 Introduction

Reinforcement learning (RL) has recently achieved impressive success in a variety of complex decision-making tasks, such as robotics (Yang et al., 2022; Wu et al., 2023) and gaming (Silver et al., 2016; Wurman et al., 2022). However, it usually takes a huge amount of trial and error for RL algorithms to learn an effective policy. This makes the application of reinforcement learning challenging.

Recent research has introduced various methods to improve sample efficiency (Schwarzer et al., 2021; 2023). Model-based methods are considered promising approaches to accelerate agent learning. Unlike model-free methods, they learn a dynamic model of the environment, also called *world model* (Ha & Schmidhuber, 2018), and allow the agent to interact in the world model to acquire more samples without touching the real environment, as one can learn in the imagination.

However, the use of a world model is not blind because the predictive accuracy and generalization are not guaranteed in complex environments (Plaat et al., 2023). The *rollout length*, which is used to limit the imaginary horizon of the agent in the world model, is a critical hyperparameter in model-based approaches (Janner et al., 2019). Intuitively, a longer rollout length leads to greater sample efficiency since more data are generated. However, as long trajectories are generated, the prediction accuracy at each step decreases due to the compounding of model error, resulting in poor generation quality. Thus, the rollout length plays a crucial role as a trade-off between data quality and efficiency, which needs to be set carefully. Previous approaches tend to achieve better performance by manually adjusting the rollout length. However, using a fixed rollout length is not optimal for all rollouts during the training process (Nguyen et al., 2018). There are some approaches that try to utilize metrics from the training process for automatic adaptation (Nguyen et al., 2018; Xiao et al., 2019) but only slightly adjusted for simple environments.

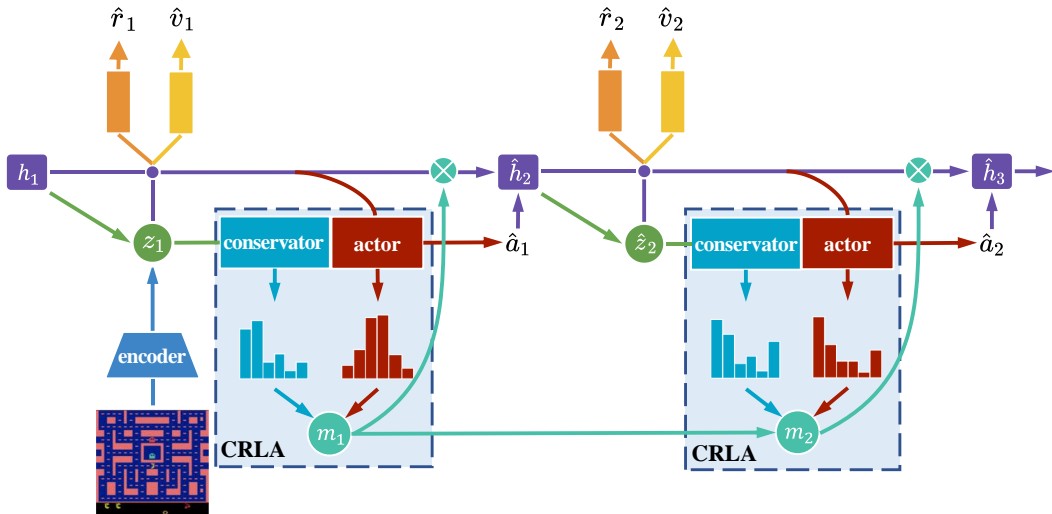

Figure 1: Overview of CRLA applied to Dreamerv3. The conservator predicts the practiced action distribution with the current latent state $z_t$. At each step of the rollout, CRLA calculates the distance between the output distributions of the actor and conservator. Then the mask $m_t$ is computed based on the distance with respect to the set threshold. If $m_t = 0$, then the subsequent rollout needs to be truncated.

Intuitively, a conservative strategy for a safe rollout is to prefer practiced actions that have already been taken frequently in the current state when imagining, since humans usually avoid imagining outside the box if they lack a comprehensive understanding of the dynamics. Based on this inspiration, we propose a novel **C**onservative **R**ollout **L**ength **A**daptation method called **CRLA**. Our main idea is that the agent, when interacting with the world model, should try to choose practiced actions in the current state for safe rollout and truncate the rollout when there is a high probability of selecting unpracticed actions that are seldom taken. The overall framework of CRLA is shown in Figure 1. We train a neural network called *conservator* to predict the distribution formed by the take frequency of actions at each state. We determine whether it is safe to continue the rollout by judging if the output distributions of the conservator and the actor are close enough to each other, and truncate the rollout if it is not safe. Our approach is a conservative rollout strategy that prevents the rollout from falling into regions with large prediction errors by truncating the rollout when there is a high probability of selecting rarely taken actions. We evaluate CRLA applied to DreamerV3 (Hafner et al., 2023) on the Atari100k benchmark. Note that our method can be applied to most model-based method that works in the discrete action space environment. CRLA demonstrates a notable performance improvement over Dreamerv3 in most Atari games, indicating its ability to effectively strike a balance between data quality and efficiency.

**Our contribution:** The contributions of this work can be summarized as follows:

1. We introduce a conservative rollout strategy that stops unrolling when the agent selects an action seldom chosen in the current state.

2. We propose a novel conservative rollout length adaptation method following this strategy, aimed at discarding potentially unrealistic transitions for safety.

3. We evaluate CRLA applied to DreamerV3 on the Atari100k benchmark and achieves a significant improvement and demonstrates that further performance gains can be achieved by dynamically adjusting the rollout length.

## 2 BACKGROUND

We consider a partially observable Markov decision process (POMDP) with discrete time steps $t \in \mathbb{N}$, high-dimensional image observations $x_t \in \mathbb{R}^{h \times w \times c}$, discrete actions $a_t \in \{1, ..., m\}$ and

scalar rewards $r_t \in \mathbb{R}$. Episode ends are indicated by a boolean variable $d_t \in \{0, 1\}$. The goal is to find a policy $\pi$ that maximizes the expected sum of discounted rewards $\mathbb{E}_\pi \left[ \sum_{t=1}^\infty \gamma^{t-1} r_t \right]$, where $\gamma \in [0, 1)$ is the discount factor.

To introduce our method more conveniently, we specify it based on the Dreamerv3 algorithm, a state-of-the-art model-based reinforcement learning algorithm (Hafner et al., 2023), since our method is a general approach which can be applied to most model-based method that works in the discrete action space environment. We briefly describe the framework of Dreamerv3 below.

**World model:** One of the fundamental components of the model-based algorithm is the world model, which learns compact representations of observations and predicts future representations and rewards with potential actions. To process high-dimensional image inputs, the world model requires an encoder that learns compact representations to encode image observations $x_t$ into hidden states $z_t$ (Kingma & Welling, 2013). Then an RNN-based sequence model predicts the next recurrent state $h_t$ based on past state $z_{t-1}$ and action $a_{t-1}$. The dynamics predictor predicts the next latent state $\hat{z}_t$ based on the recurrent state. These three modules form the Recurrent State-Space Model (RSSM) (Hafner et al., 2019), which is the core of the world model:

$$\text{RSSM} \begin{cases} \text{Sequence model:} & h_t = f_\phi \left( h_{t-1}, z_{t-1}, a_{t-1} \right) \\ \text{Encoder:} & z_t \sim q_\phi \left( z_t \mid h_t, x_t \right) \\ \text{Dynamics predictor:} & \hat{z}_t \sim p_\phi \left( \hat{z}_t \mid h_t \right) \end{cases} \quad (1)$$

The concatenation of the hidden state and the current state is used to predict the reward $r_t$, the episodic continuation flags $c_t \in \{0, 1\}$ and the next observation $x_t$ for learning compact representations:

$$\begin{array}{ll} \text{Reward predictor:} & \hat{r}_t \sim p_\phi \left( \hat{r}_t \mid h_t, z_t \right) \\ \text{Continue predictor:} & \hat{c}_t \sim p_\phi \left( \hat{c}_t \mid h_t, z_t \right) \\ \text{Decoder:} & \hat{x}_t \sim p_\phi \left( \hat{x}_t \mid h_t, z_t \right) \end{array} \quad (2)$$

**Actor-Critic learning:** Dreamer uses actor-critic framework for policy optimization. Both actor $\pi_\theta(a_t|s_t)$ and critic $v_\psi(s_t)$ operate on model states $s_t = \{h_t, z_t\}$ and are trained on-policy entirely on trajectories imagined by world model. Real trajectories are sampled from the replay buffer and used as starting points to generate imaginary trajectories in length $T$. We then compute bootstrapped $\lambda$-returns (Sutton & Barto, 2018) on these trajectories and use this for optimization:

$$R_t^\lambda = r_t + \gamma c_t \left( (1 - \lambda) v_\psi \left( s_{t+1} \right) + \lambda R_{t+1}^\lambda \right) \quad R_T^\lambda = v_\psi \left( s_T \right) \quad (3)$$

## 3 METHOD

In this section, we first explain the idea of the conservative rollout strategy. Then, we introduce our method inspired by this strategy and provide a practical implementation based on Dreamerv3. Finally, we illustrate the theoretical support behind our method.

### 3.1 IMAGINE WITHIN PRACTICE

Due to the limitation of interaction steps, it is difficult for the world model to fully and accurately capture the dynamics of the environment. However, during the rollout process, it is essential for the world model to possess the capability of generalization to produce novel transitions that have not been previously observed. This generalization alone cannot ensure the quality of imagined transitions, as the world model learns only from limited data. Since the model error cannot be eliminated, and even minor errors can be compounded by multi-step rollout, this has a detrimental effect on policy optimization.

Intuitively, since the world model is trained with practiced trajectories in the replay buffer, it can predict future information more accurately when it encounters frequently seen transitions. When the agent chooses an action that has not been practiced while interacting with the world model, the world model tends to produce large generalization error. Figure 2 illustrates this situation. Selecting unpracticed actions is unsafe as it can lead the rollout to deviate from the accurate prediction region and introduce risk to subsequent rollouts due to the compounding of model error.

To ensure the safety of the rollout, a conservative strategy is to make the agent imagine within practice, since the world model is trained with practiced trajectories. This means that we want the agent to be able to determine whether the current action choice has been practiced sufficiently to make confident predictions during the imagination process. If the chosen action has rarely been practiced in the current state, the imagination should be interrupted to avoid falling into unrealistic fantasies. If the agent has a high probability of choosing an action that has rarely been practiced in the current state, the imagination should be interrupted so as not to fall into unrealistic fantasies.

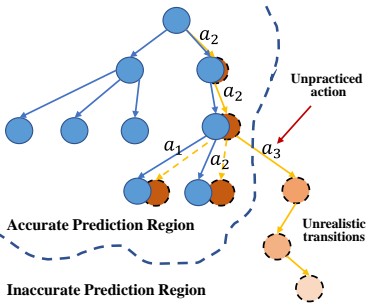

Figure 2: Illustration of the conservative rollout. The blue circles and arrows in the figure represent real states and transitions, respectively, and the orange ones are imagined. If the agent chooses practiced actions, this results in a smaller prediction error. While, if it chooses unpracticed actions, it may deviate the state out of the region where the world model predicts accurately, leading to the generation of unrealistic trajectories.

### 3.2 CONSERVATIVE ROLLOUT LENGTH ADAPTATION

Based on this intuitive strategy, we propose a novel conservative rollout length adaptation method called CRLA. Specifically, we define $\pi_p(a_t \mid z_t)$ to represent the practiced action distribution, which is shaped by the frequency of each action taken at each latent state. Note that $\pi_p$ does not use $s_t = \{h_t, z_t\}$ as input, since it does not need to consider the context $h_t$. With $\pi_p$, we can identify which actions have been taken in the current latent state and their frequency to determine whether the action is practiced enough or not. Our main idea is to truncate the rollout when the agent has a higher probability of choosing actions that are not sufficiently practiced. This means that the rollout will continue only when the distance between $\pi_p$ and $\pi_\theta$ is sufficiently small. In this paper, we employ the Jensen–Shannon divergence as our distance metric. For each rollout step, we calculate the distance between $\pi_p$ and $\pi_\theta$ respectively for each rollouts. We set a threshold $\alpha$ to determine whether to continue the rollout. If the distance is less than $\alpha$, the rollout will continue, otherwise subsequent rollouts will be masked with $m_t$, as shown in Equation 4. Note that our approach does not directly judge the final action selection, but instead uses the action distribution as the basis for judgment, which means that the agent still has the probability to sample the untaken actions when the constraint is satisfied, preserving the exploration ability of the agent.

However, $\pi_p$ is not easy to acquire. An easily thought of method is to directly count the frequency of practiced actions at each state in the replay buffer. But there are several problems with this. First, directly counting the frequency of practiced actions under the original observation is convenient but not reasonable, as the prediction of the world model is made in the latent space. Even if the observations are not identical, they may map to the same point in the latent space. Second, counting within the latent space would require re-counting after the update of the encoder, which significantly increases the computational cost. Lastly, owing to the world model's generalization capacity, it generates latent states that do not truly exist but are close to the real one. These problems make it infeasible to obtain $\pi_p$ by counting.

$$m_t = \begin{cases} 1, & \text{if } D_{JS}(\pi_D(z_t), \pi_\theta(s_t)) < \alpha \text{ and } m_{t-1} = 1 \\ 0, & \text{if } D_{JS}(\pi_D(z_t), \pi_\theta(s_t)) \geq \alpha \text{ or } m_{t-1} = 1 \end{cases} \tag{4}$$

In order to efficiently acquire an approximation of $\pi_p$, we parameterized it using a neural network. We call it *conservator* and use $\hat{\pi}_p(a_t \mid z_t)$ to denote. The conservator takes latent states $z_t$ as input and outputs the practiced action distribution under $z_t$. We train it using practiced trajectories from the replay buffer. Since one latent state can correspond to multiple actions, we refer to multi-label learning approaches. However, in the multi-label classification task, each input necessitates the provision of the complete label set for supervised training. This presents a challenge to our task because it requires iterating through the entire replay buffer to find all actions corresponding to each latent state, resulting in a significant increase in computational complexity. Therefore we use sampling for training, where the state-action pairs are uniformly sampled from the replay buffer at each training step. We then embed the original observations into latent states as input and use the one-hot coding of the actions as labels. We utilize the binary cross entropy loss function that is often

used in multi-label learning tasks to calculate the loss shown in Equation 5. We normalize the final output logit of the conservator to obtain the predicted distributions of practiced actions. Because the sampling is uniform, the conservator is able to capture the frequency of each action in each latent state to approximate the $\pi_p$. We validate this on the mnist dataset, referring to the Appendix A.2 for detailed results.

$$\mathcal{L}_{BCE}(\hat{\mathbf{y}}, \mathbf{y}) = -\frac{1}{N} \sum_{i=1}^{N} y_i \cdot \log(\hat{y}_i) + (1 - y_i) \cdot \log(1 - p(\hat{y}_i)) \tag{5}$$

$$\mathcal{L}_{statistic} = \mathop{\mathbb{E}}_{s,a \in D}[\mathcal{L}_{BCE}(\hat{\pi}_D(encoder(z \mid s)), one\_hot(a)] \tag{6}$$

After obtaining $\hat{\pi}_p$, we can calculate the Jensen–Shannon divergence between $\hat{\pi}_p(a_t \mid z_t)$ and current policy $\pi_\theta(a_t \mid s_t)$ at each rollout step in the world model, and determine whether to continue unrolling. For computational convenience, we first compute the trajectory with the max rollout length, and then apply the mask $m_t$ to mask out those states that do not satisfy the condition and their successors. When calculating the bootstrapped $\lambda$-returns, it should be calculated respectively for different rollout lengths according to the mask as shown in Equation 7. $F_t \in \{0, 1\}$ is the flag of the first invalid state which is equal to 1 only in this case to truncate the bootstrap returns of the invalid state.

$$R_t^\lambda = m_t \left[ r_t + \gamma c_t \left( (1 - \lambda)v_\psi(s_{t+1}) + \lambda(R_{t+1}^\lambda \cdot (1 - F_t) + v_\psi(s_{t+1}) \cdot F_t) \right) \right] \tag{7}$$

The setting of the threshold $\alpha$ is crucial for our method. To simplify the design of the threshold, we would like to set one threshold for all 26 Atari games. However, since the dimensions of action vary across Atari games, it may not be appropriate to set only one threshold for them. The reason is that for environments with small action dimensions, relatively low thresholds need to be set to provide a sensitive truncation of the rollout. For environments with large action dimensions, however, relatively high thresholds need to be set to avoid overly strict judgment conditions. Therefore, we use a threshold adjustment approach that is adaptive to the dimensionality of the action. We define $p, q$ as two different n-dimensional one-hot vectors and $u$ as an n-dimensional uniform distribution. We set a hyperparameter $\beta$ and compute the threshold $\alpha$ using the following equation:

$$\alpha = D_{JS}[(\beta p + (1 - \beta)u), (\beta q + (1 - \beta)u)] \tag{8}$$

In summary, our method has the following advantages: (1) Adaptation. Our method adapts the rollout length for each rollout individually, thus utilizing the world model as much as possible while still being safe. (2) flexibility. Since the Jensen–Shannon divergence between $\hat{\pi}_p$ and $\pi_\theta$ is used as the judgment condition, there remains a possibility for the agent to explore unpracticed actions while adhering to the constraints. Our method can be regarded as a form of soft rollout constraint.

### 3.3 THEORETICAL ANALYSIS

Previous researches have conducted theoretical analyses of the gap between returns under a branched rollout scheme and real environment interactions, and derived a bound. The branched rollout scheme is that we begin a rollout from a state under the previous policy's state distribution $d_{\pi_D}(s)$ and run k steps according to current $\pi$ under the learned world model $p_\theta$. We analyze the validity of our method based on this basis.

**Theorem 3.1 (Janner et al., 2019)** *Let the expected total variation between two the learned model is bounded at each timestep under the expectation of $\pi$ by $\max_t E_{s \sim \pi_t} [D_{TV}(p(s' \mid s, a) \| \hat{p}(s' \mid s, a))] \leq \epsilon_{m'}$, and the policy divergences are bounded as $\max_s D_{TV}(\pi_D(a \mid s) \| \pi(a \mid s)) \leq \epsilon_\pi$, where the $\pi_D(a \mid s)$ denote the data-collecting policy. Let the $\eta[\pi]$ denotes the returns of the policy in the true MDP. Then under a branched rollouts scheme with a branch length of k, the returns $\eta^{\mathrm{branch}}[\pi]$ are bounded as:*

$$\eta[\pi] \geq \eta^{\mathrm{branch}}[\pi] - 2r_{\max} \left[ \frac{\gamma^{k+1}\epsilon_\pi}{(1 - \gamma)^2} + \frac{\gamma^k \epsilon_\pi}{(1 - \gamma)} + \frac{k}{1 - \gamma} (\epsilon_{m'}) \right] \tag{9}$$

To reduce the gap between $\eta^{\mathrm{branch}}[\pi]$ and $\eta[\pi]$, we need to reduce the second term on the right-hand side of Equation 9 as much as possible. In this item, there are three key factors: the model error

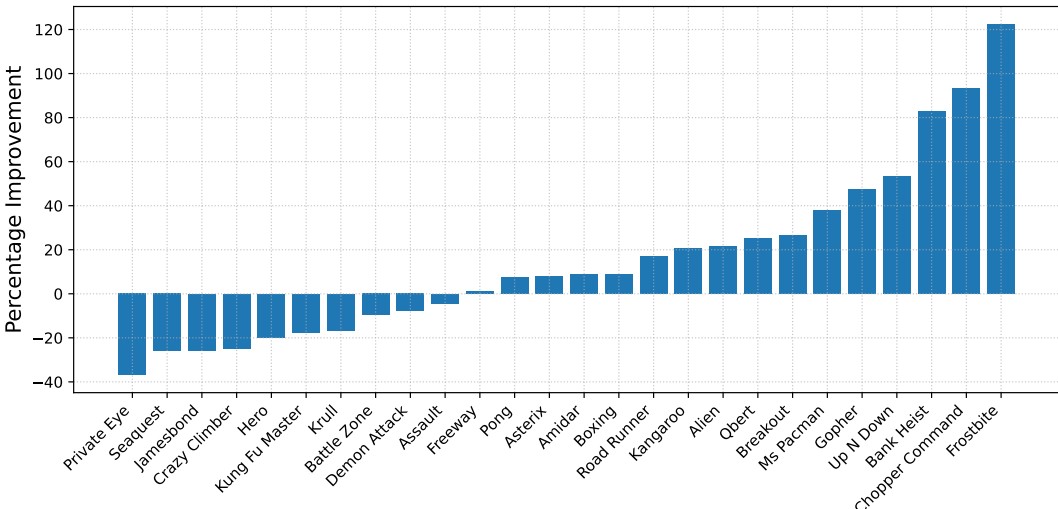

Figure 3: Improvement in percentage of Dreamerv3-CRLA over default Dreamerv3. We perform 10 runs per game and compute the average over 100 episodes at the end of training for each run.

$\epsilon_{m'}$ under the current policy, the policy distribution shift $\epsilon_\pi$ between the current policy $\pi$ and the data-collecting policy $\pi_D$ and the rollout length $k$. For the model error $\epsilon_{m'}$, since the world model is trained by supervised learning to fit the data in the replay buffer rather than the full dynamics of the real environment, this leads to the generalization error. The error can be substantial for transitions that are seldom observed. Our method expects the agent to try to select practiced actions that have been taken before in the current state when interacting with the world model. We truncate the rollout when the agent chooses an unpracticed action, as this can cause large model errors. This avoids a further increase in the term $\frac{k}{1-\gamma}\left(\epsilon_{m'}\right)$, which increases with rollout length $k$. For the policy distribution shift $\epsilon_\pi$, since the *conservator* is the approximation of $\pi_D(a \mid s)$, our method actually constrains $\epsilon_\pi$ explicitly. We add constraints to the rollout so that it continues when the policy's action distribution $\pi_\theta(a \mid s)$ is similar to $\pi_D(a \mid s)$. This allows policy distribution shifts $\epsilon_\pi$ to be restrained during rollout thus protecting the quality of the generated trajectories. By dynamically adjusting the rollout length $k$ with our method, the model error and the policy distribution shift can be effectively constrained. This theoretically supports our method.

## 4 EXPERIMENT

In this section, we aim to answer the following questions: (1) Whether CRLA can improve performance by adjusting only the rollout length? (2) Whether CRLA can balance data quality and efficiency? (3) Whether CRLA can truncate the rollout at the appropriate step? (4) For what kind of environments CRLA causes performance degradation?

To answer these questions, We evaluate CRLA applied to DreamerV3 on the Atari100k benchmark. The Atari 100k benchmark (Kaiser et al., 2020) includes 26 games from the Arcade Learning Environment (Bellemare et al., 2013) and the agent is only allowed 100,000 steps of environment interaction per game, which are 400,000 frames with a frame-skip of 4 and corresponds to roughly two hours of real-time gameplay. It can effectively test the sample efficiency of the method in case of limited interactions. We want to test whether CRLA can effectively truncate those harmful generated trajectories when the interaction steps are limited. To avoid tedious hyperparameter tuning, we set $\beta = 0.78$ to automatically calculate thresholds $\alpha$ for all 26 games in the Atari 100k benchmark. All hyperparameters are identical to the Dreamerv3 default settings, except for the rollout length. We restrict the adjustment range of the rollout length to $[5, 16]$ to avoid too long or too short rollout length, while the default setting is a fixed one $T = 15$. Due to the lack of data in the replay buffer and the instability of the encoder at the early stage, we train and apply the conservator after 10k steps to avoid overfitting. We perform 10 runs per game and compute the average score over 100 episodes at the end of training for each run.

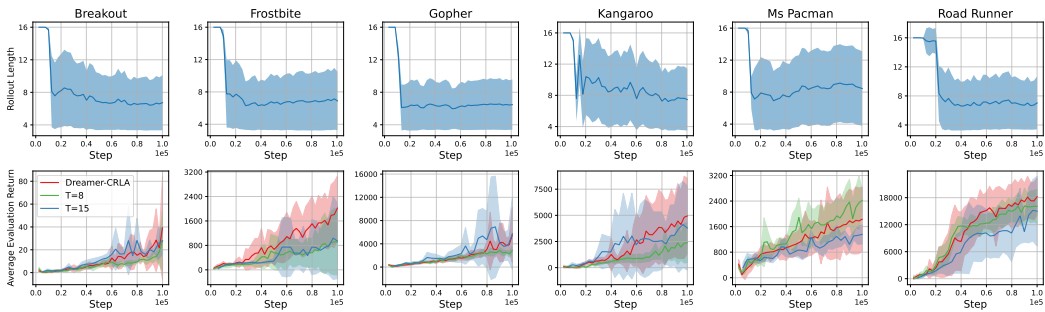

Figure 4: The rollout length adaptation by CRLA in six games is shown in the first line and the comparison of the different rollout length in each game is shown in the second line. The solid line is the mean over 10 seeds for our method and 5 seeds for Dreamerv3 with $T = 8$ and $T = 15$, while the shaded area represents one pointwise standard deviation.

## 4.1 PERFORMANCE IMPROVEMENT

We aim to assess whether CRLA could improve the performance of Dreamerv3 by adjusting the rollout length. Figure 3 illustrates the percentage performance improvements across all 26 Atari games compared to default Dreamerv3 with a fixed rollout length $T = 15$. It shows that CRLA significantly improves the performance of Dreamerv3 in most games. See the Appendix A.1 for detailed training curves. The results sufficiently demonstrate the effectiveness of CRLA. It's noteworthy that we did not individually tune the threshold $\alpha$ for each game, emphasizing the user-friendliness of CRLA. And we believe that positive performance improvements can be achieved by fine-tuning the threshold and the adjustment range for each game respectively, since we only adjust the rollout length and do not modify the other hyperparameters.

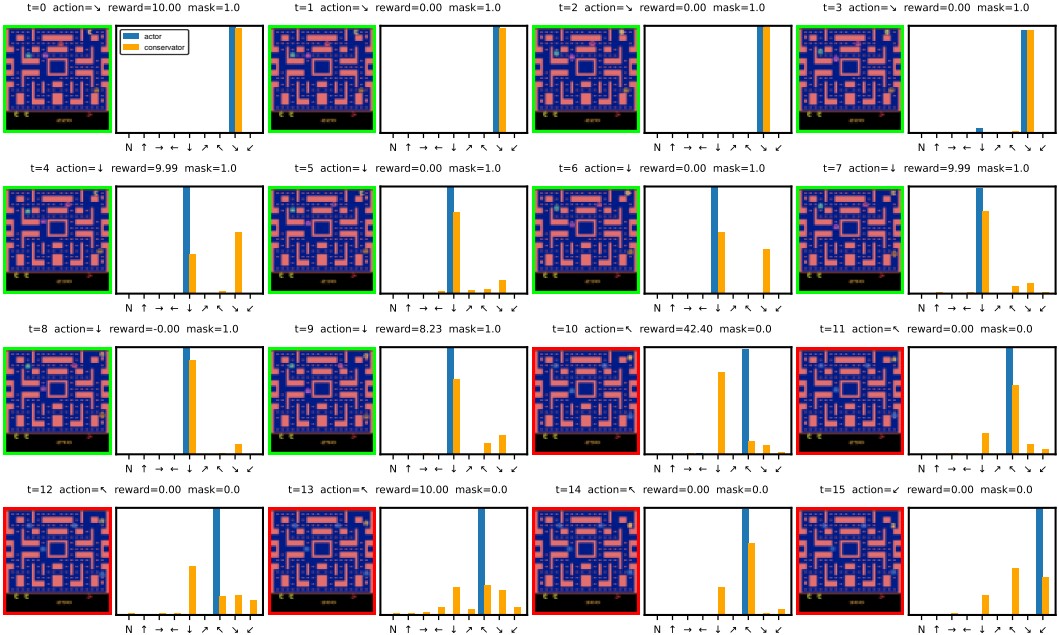

Figure 5: Full rollout trajectory on Ms Pacman. The image at $t = 0$ is the real observation as a starting point. The player controls the yellow Pac-Man in the upper right corner of the image at $t = 0$. The green border of the image represents $m_t = 1$ and the red border represents $m_t = 0$. The bars on the right side of each image show the action distribution output by the policy with blue bars and the practiced action distribution output by the conservator with yellow bars ranging from 0 to 1.

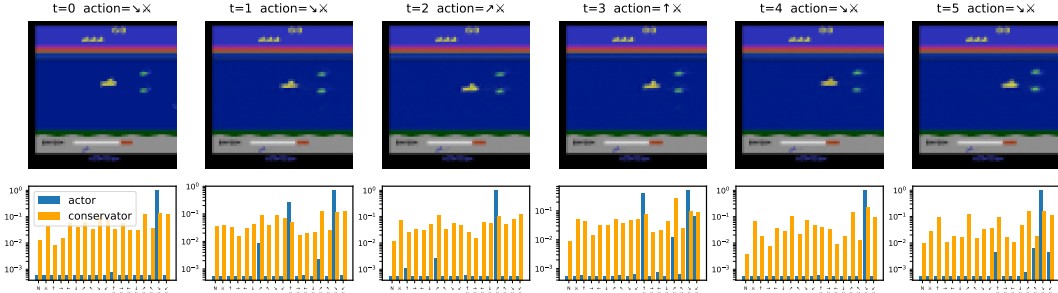

Figure 6: Partial rollout trajectory on Seaquest game. The bars below each image correspond to the output distributions of the policy with blue bars and the conservator with yellow bars at each step.

## 4.2 BALANCING DATA QUALITY AND EFFICIENCY

As each rollout varies in length, we record the mean and variance of the rollout length in the batch data at each training step, as shown in the first row of Figure 4. We aim to investigate whether our approach can achieve greater efficiency compared to the fixed rollout length setup when using a similar number of imagined rollout transitions. For convenience, we set the fixed rollout length $T = 8$, which closely approximates the average rollout length in our method. The results of this comparison are presented in the second row of Figure 4. As $T$ varies from 15 to 8, it results in a performance degradation at the fixed rollout length setup in some games. However, Our method is more sample efficient than both of them while using a smaller number of imagined rollout transitions but achieving better performance. It can be seen that the average rollout length stays in the middle region of the set range with a large variance to cover the entire range in our method. This suggests that CRLA has the ability to safely and flexibly adjust the rollout length to balance data quality and efficiency.

## 4.3 ANALYZING THE VALIDITY OF TRUNCATION

We want to observe where CRLA would choose to truncate the rollout. For illustration, We visualize the rollout trajectory of the Atari game Ms-Pacman. To demonstrate the validity of the conservator, we visualize the actor's output distribution and the conservator's output distribution at each step in the rollout trajectory. Figure 5 presents comprehensive information about the entire rollout trajectory. We can see that CLRA chooses to terminate the rollout at $t = 10$ when the action distribution significantly deviates from the output of the conservator. At this step, the agent selected the action UPLEFT, predicted by the conservator to be a rarely practiced action, while the action DOWN was considered a frequently practiced action. From $t = 11$ to 14, the agent selected the action UPLEFT, but the reconstructed observations reveal that the agent actually moved down. This illustrates that the world model is overfitted to the action DOWN at this latent states, resulting in an incorrect rollout trajectory. This addresses the third question and demonstrates that CRLA can effectively truncate rollouts at critical points. Appendix A.3 shows the more rollout trajectories of some other games.

## 4.4 ANALYSIS OF PERFORMANCE DEGRADATION

Based on our experimental results, we would like to explore under which scenarios CRLA may lead to performance degradation. We select the Seaquest game as an illustrative example, since it shows performance degradation after using CRLA. In Figure 7, we present a partial rollout trajectory of the Seaquest game. A notable difference compared to the Ms-Pacman game is that the *conservator*'s output is more uniform in Seaquest, indicating that it predicts many actions in the current state with similar frequencies. In contrast, the actor is more explicit in its decisions, with a high probability to select a certain action. There are two possible reasons for the more uniform output of the conservator, one is that the action selection is more uniform when interacting with the environment for sampling, and the other is that the encoding of the observation in the latent space is not well learned, leading to latent state confusions. In this case, it may be difficult for the conservator to capture the real practiced action distribution. This causes conservator's judgment becomes very sensitive to the

threshold $\alpha$. The larger action space can cause this problem as well. In this case, the threshold need to be set carefully.

## 5 RELATED WORK

Model-based reinforcement learning methods improve sample efficiency by interacting with the learned world model. However, the model errors prevent model-based approaches from acquiring high-quality data from the world model. Previous works have found that even small model errors can be compounded by multi-step rollout and deviate the predicted state from the region where the model has high accuracy.

To mitigate the effects of compounding error, previous work has proposed many improvements. Some methods improve the model to achieve more accurate predictions. Kaiser et al. (2020) reduced prediction complexity by embedding complex high-dimensional image observations into low-dimensional hidden spaces using deep convolutional neural networks. Hafner et al. (2019) proposed the Recurrent State Space Model (RSSM) and achieved outstanding prediction accuracy. Micheli et al. (2022) and Robine et al. (2023) utilized the powerful sequence modeling capabilities of transformer to accurately learn the dynamics of the environment. Other approaches reduce the model error by improving the training scheme. Yu et al. (2021) introduced the cycle-consistency constraint for representation and model learning to improve the sample efficiency. Eysenbach et al. (2022) proposed a single objective for jointly training the model and the policy to tackle the objective mismatch problem. Ghugare et al. (2022) designed aligned latent models to simplify the training of the latent-space model and policy and remain self-consistent.

However, since it is difficult to fully explore the whole state space in complex environments, the error of the world model cannot be completely eliminated. One idea is to mitigate the effects of model error by limiting the rollout length. Nguyen et al. (2018) argued that the fixed rollout length was problematic and proposed an adaptive rollout method using uncertainty estimation but only for simple deterministic environments. Xiao et al. (2019) introduce adaptive model-based value expansion method that adaptively selects planning horizons for each state according to the estimated compounding error but still can only plan horizons in a small range. Lai et al. (2020) develop bidirectional models to generate trajectories in the forward and backward directions at the starting point to reduce the compounding error without decreasing the rollout length. However, it still used a fixed rollout length. Lai et al. (2021) utilized metrics from the training process to guide rollout length adjustment but required additional training data to train the hyper-controller.

Our approach differs from previous methods in that we introduce a conservative strategy to adjust the rollout length rather than utilizing metrics from the training process such as training loss. Our method is computationally simpler compared to previous methods and can safely and efficiently adjust the rollout length to balance data quality and efficiency.

## 6 CONCLUSION AND DISCUSSION

In this paper, we propose a novel conservative rollout length adaptation method called CRLA, which prevents the rollout from falling into regions with large prediction errors by truncating the rollout when there is a high probability of selecting rarely taken action. CRLA avoids the rollout trajectory that deviate too far from the true transition by conservatively truncating the rollouts. We validate the effectiveness of our method through experimental results and theoretical analysis. We evaluate CRLA applied to DreamerV3 on the Atari100k benchmark and achieve significant performance gains in most environments. We believe that our work is an important step towards further improving the performance of model-based reinforcement learning methods. The limitations of our work are that it is only applicable to the discrete action space and the generalization of the conservator may not be sufficient since it is trained only on real samples but needs to be evaluated on imaginary trajectories. We will look into this further in our future work.

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

# A APPENDIX

## A.1 DETAILED TRAINING CURVE

Figure 1 shows the full training curve of our method compared to the original Dreamerv3. Dreamerv3's training data is obtained from the official code `https://github.com/danijar/dreamerv3`.

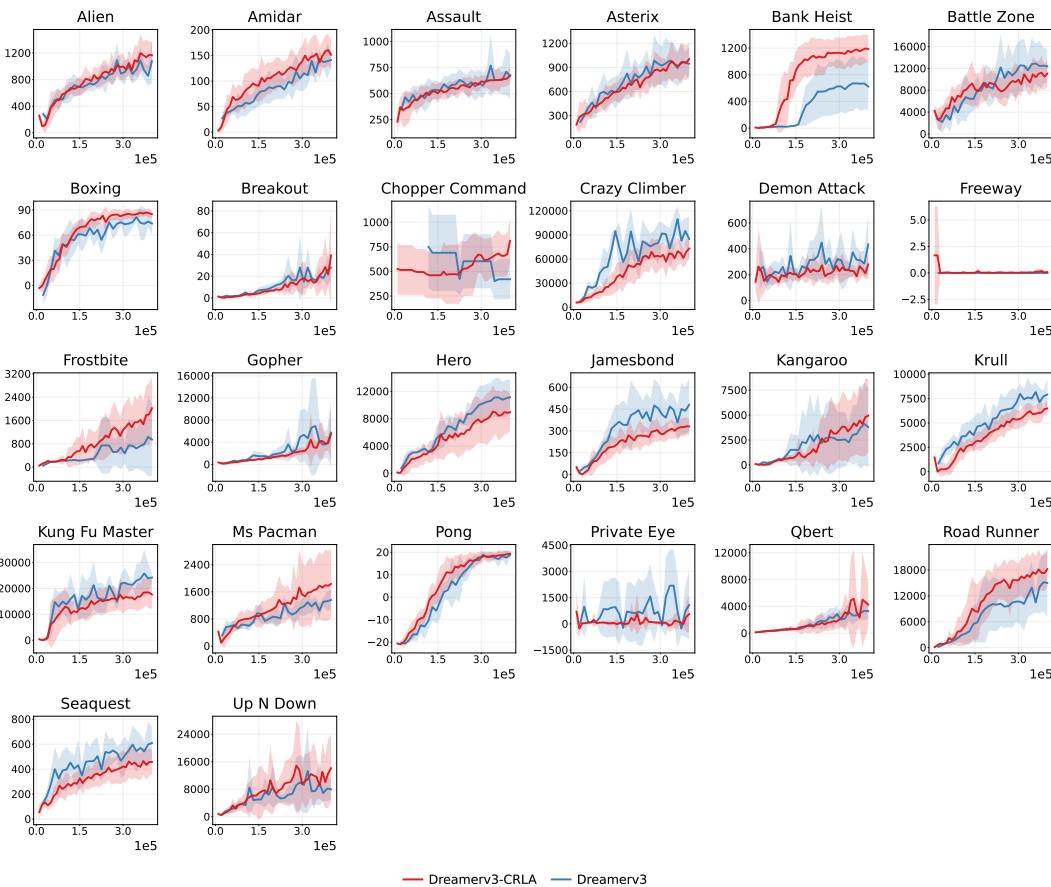

Figure 7: Learning curves for DreamerV3 with and without CRLA on the 26 environments of the Atari 100k benchmark.The solid line is the mean over 10 seeds for our method and 5 seeds for Dreamerv3, while the shaded area represents one pointwise standard deviation.

## A.2 THE VALIDITY OF CONSERVATOR'S TRAINING

To verify that the training of the conservator is effective, we conduct validation experiments on the mnist dataset. To simulate the case where one state corresponds to multiple actions, we randomly modify the labels of each image. We specify a label distribution for each class of images and randomly sample labels from this distribution to override the original labels. We want to investigate whether the sampling-based training approach allows the conservator to capture the corresponding label distribution under each class of images. We simulate the computational flow in training by using an autoencoder to compress the image into the latent space and learn the predicted label distribution in the latent space. Figure 8 shows that the real label distribution can be captured by sampling, which proves the validity of our method.

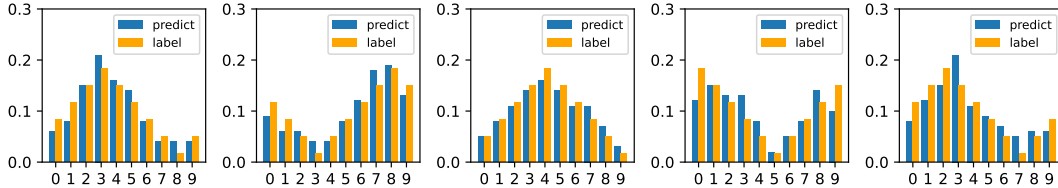

Figure 8: The fitting results of the label distribution. We design specific label distributions for each class of images and randomly sampled from them to replace the original labels. We train 10 epochs with sampling and show the evaluation of the predictive distribution to the label distribution.

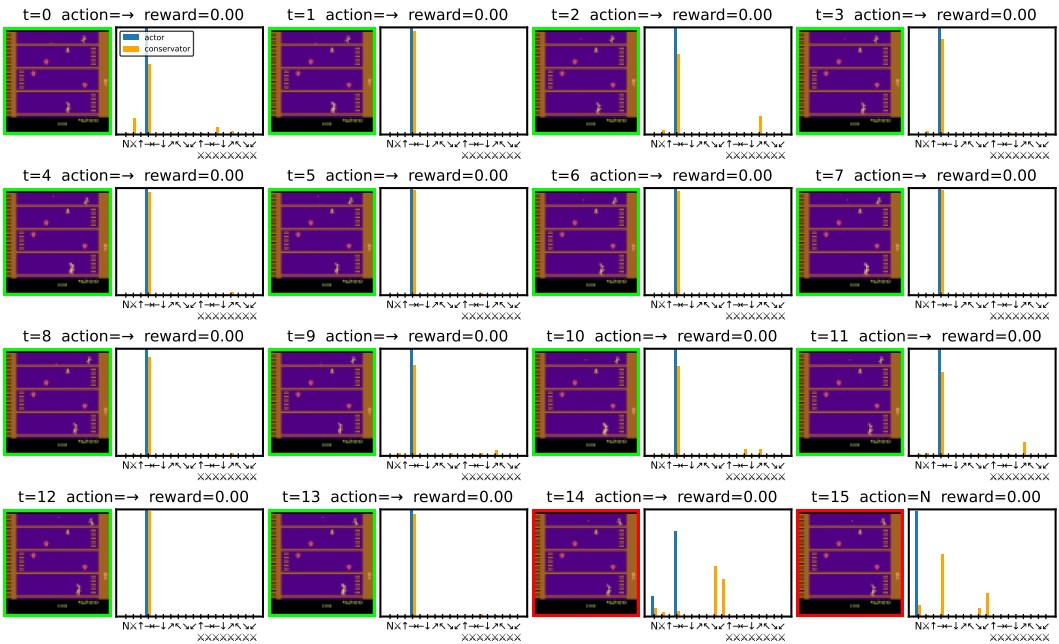

Figure 9: Full rollout trajectory on Kangaroo game. The player controls the action of the yellow villain at the bottom.

## A.3 EXTENDED EXPERIMENTS

We provide more example trajectories in Figure 9 and Figure 10. On both games, our approach achieves performance improvement. As can be seen from the rollout trajectory, the conservator is able to accurately determine which actions are frequently practiced and truncate the rollout at critical moments when the agent selects an unpracticed action. It can be seen from the reconstructed observations and action choices that although the world model can predict accurately on certain unpracticed actions due to its generalization ability, this can be risky, which is why our approach is a conservative strategy.

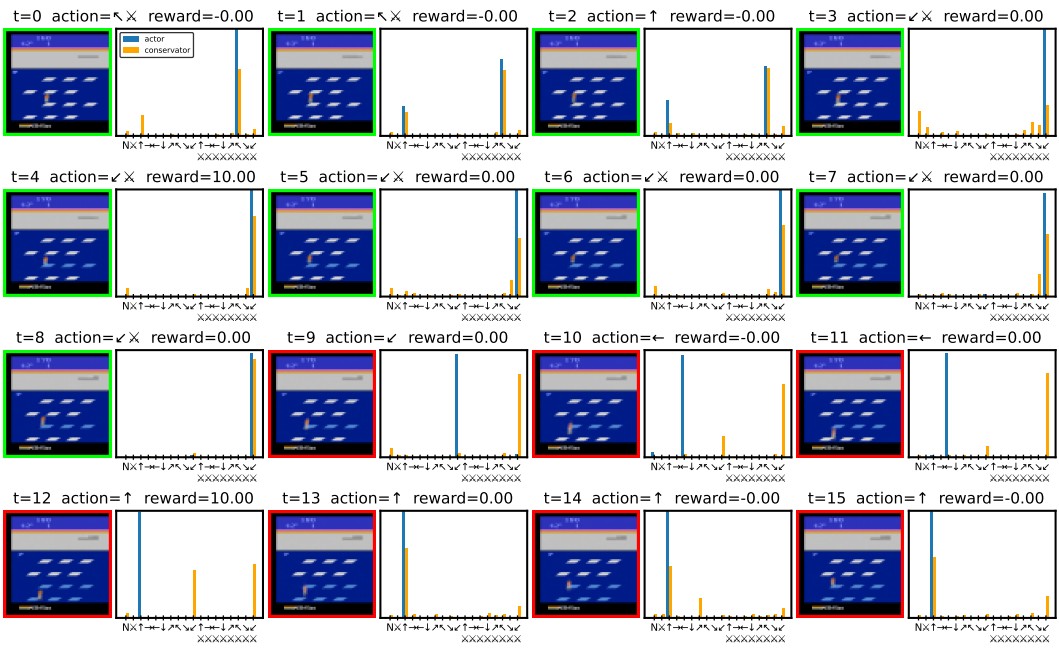

Figure 10: Full rollout trajectory on Frostbite game. The player controls the action of the villain on the white platform.

