# OpenReview forum: "Imagine Within Practice: Conservative Rollout Length Adaptation for Model-Based Reinforcement Learning"
_ICLR.cc/2024/Conference — Submitted to ICLR 2024_

### Official Review · Reviewer_dUDY · 2023-10-27

**Soundness:** 2 fair
**Presentation:** 3 good
**Contribution:** 1 poor
**Rating:** 5
**Confidence:** 4

**Summary:**

This paper proposed a model-based RL method, called CRLA, to tune the rollout length in model-based RL settings adaptively. To achieve this, CRLA truncates the rollout while the agents tend to select an infrequently taken action. Then, the authors introduced  CRLA to dreamerV3 and conducted several experiments on the Atari 100 benchmark. Some empirical results in the Atari 100k benchmark demonstrated the effectiveness of CRLA.

**Strengths:**

1. This paper is well-written and easy to understand.
2. The method is concise and clear, and the proposed problem is indeed one of the challenges in current Model-Based RL.

**Weaknesses:**

1. Figure 4 shows that the rollout length will generally decrease in the late training stage in most tasks. It can be seen here that CRLA seems to be able to avoid excessively long rollout, but it does not seem easy to see whether CRLA can adaptively trade-off the rollout since few experiments show that CRLA knows when to increase the trajectory length.
2. In the experiment in Figure 4, CRLA achieves performance beyond fixed length on only about half of the tasks. This may be not enough to support the effectiveness and significance of CRLA.
3. The author does not appear to have provided the code, so reproducibility may be difficult to guarantee.

**Questions:**

1. While CRLA early stops those harmful trajectories, will some promising trajectories be truncated as well? For example, in Ms Pacman, CRLA seems to be outperformed by some fixed rollout length methods.
2. CRLA may be able to avoid the cumulative errors caused by longer trajectories, but CRLA may also cause some good exploration trajectories to be terminated early, which may be harmful. Hopefully, the author can alleviate this concern.

---

> ### Author Response · Authors · 2023-11-13
>
> Thank you for your helpful comments on our paper!
> * **Weakness1**: It is difficult to explicitly show experimentally when CRLA increases trajectory length.  We attempt to illustrate how CRLA works and its effectiveness using visualization results in 4.3. Figure 5 shows how CRLA works by truncating the trajectory at key time steps, suggesting that our method is able to determine when truncation is necessary. But it is difficult to show under what conditions CRLA is favored to increase the rollout length, since our method is a conservative method. Figure 4 shows the average step size of the unfolding during training, but does not show the maximum length in the trajectories. In fact, in our experiments, trajectories reaching the maximum rollout length existed on almost all environments at any period of training.
> * **Weakness2**: Although our algorithm still shows performance degradation on 12 games, this is due to the fact that we use an automatic threshold setting method which makes it only necessary to set the same threshold parameter \\(\\beta\\) for all 26 environments. Our method does not change the policy update rules, but only adjusts the rollout length.  We could adjust the thresholds individually for those environments where performance degradation occurs and be able to guarantee that no performance degradation occurs (by simply narrowing the adjustment range so that it is closer to 15), but this does not seem elegant enough. Our method uses the same parameter \\(\\beta\\) to achieve a boost in 16 out of 26 Atari games, which we believe reflects the robustness of our approach.
> * **Weakness2**: Our approach is very easy to implement based on Dreamerv3. Our code is based on (dreamerv3-torch)[https://github.com/NM512/dreamerv3-torch] for development and we are currently organizing the code. We will submit the code in a few days.
> * **Question1**: We believe that CRLA, as a conservative rollout strategy, does have the potential to truncate promising trajectories. While World models do generate some well-generalized trajectories, there is still the possibility of early truncation. This can be adjusted by setting the threshold \\(\\alpha\\).  Although our approach is a conservative strategy, since the judgment condition is the js-divergence of the actor and conservator‘s outputs, there is still a chance that the actor will not be truncated even if it has a probability of sampling unpracticed actions. In other words, the actor still has the ability to explore.
> * **Question2**: It is difficult to balance sample efficiency and sample quality very precisely. This is still an open problem in the MBRL field. CRLA does have the potential to cause some good exploration trajectories to be terminated early, but it does not completely eliminate the possibility of exploration.  As answered in question 1, although our approach is a conservative strategy, it is actually a soft constraint rather than a hard constraint since the judgment condition is the js-divergence. As long as the js-divergence is within the threshold range, it is considered safe for the current step to rollout, at which point the actor still has the probability of sampling actions that have never been practiced. This gives our approach flexibility and preserves exploration capabilities to some extent. We believe that this concern can be alleviated as long as the ability to explore is preserved.

---

> > ### Comment · Reviewer_dUDY · 2023-11-21
> >
> > Thank you for your comprehensive feedback. I carefully read other Reviewer's comments. I will keep my score until the concerns of Reviewer pvE2 and Reviewer stfm can be addressed.

---

### Official Review · Reviewer_gVqn · 2023-10-30

**Soundness:** 2 fair
**Presentation:** 3 good
**Contribution:** 2 fair
**Rating:** 3
**Confidence:** 4

**Summary:**

This paper aims to dynamically adjust a crucial hyperparameter in model-based RL, the rollout horizon. They introduce the Conservative Rollout Length Adaptation (CRLA), which trains a policy network to predict the practiced action distribution, thereby limiting the agent's choice of out-of-distribution actions during the rollout process and dynamically adjusting the rollout horizon according to the distance between the conservator's prediction and the current policy's prediction. Experimental results on the Atari environment indicate that CRLA can outperform Dreamerv3 on several tasks.

**Strengths:**

1. Dynamically adjusting the rollout horizon is a very interesting and crucial topic for model-based RL. This can significantly improve the time efficiency of MBRL algorithms.
2. This paper is well-written and easy to follow.

**Weaknesses:**

1. Whether in state-based model-based RL or visual-based model-based RL, linearly increasing the rollout horizon with the increase in environment steps has already become a standard practice (MBPO [1], TDMPC [2]). This is not as the paper claims in the abstract, ''To prevent significant deviations between imagined rollouts and real transitions, most model-based methods manually tune a fixed rollout length for the entire training process.'' Therefore, I believe the paper should provide experimental results and analysis on DreamerV3 with a linearly adjusted rollout horizon. If the performance is good, I don't see the necessity of dynamically adjusting the rollout horizon since this requires training a conservator, which will bring additional time and computational resource consumption.

2. In state-based MBRL, there are already many methods that have attempted to address the issue of model compounding error caused by an overly long rollout horizon. These methods can be directly applied to visual-based MBRL, such as discarding samples with excessively large errors (M2AC [3]) and learning a world model that is more accurate for the current policy (PDML [4]). I believe these should all be baselines for this paper, yet the paper only provides the original results from DreamerV3.

3. One advantage of the latent world model is that there is no need to explicitly predict the next observation. Instead, transitions are performed in the latent space, which to some extent avoids model error. In the abstract, this paper claims, "While longer rollout length offers enhanced efficiency, it introduces more unrealistic data due to compounding error, potentially leading to catastrophic performance deterioration." However, no references are cited, and no experimental evidence is provided. This weakens the motivation of the paper.

4. The method proposed in this paper can only be used in discrete action environments (Atari) and is not applicable to continuous control tasks, which significantly limits the practicality of the method.

5. Based on Figure 7, the method proposed in this paper does not show significant performance improvement, and in many environments, it even harms performance.

Reference:

[1] Janner et. al. "When to Trust Your Model: Model-Based Policy Optimization," in NeurIPS 2019

[2] Hansen et. al. "Temporal Difference Learning for Model Predictive Control," in ICML 2022

[3] Pan et. al. "Trust the Model When It Is Confident: Masked Model-based Actor-Critic," in NeurIPS 2020

[4] Wang et. al. "Live in the Moment: Learning Dynamics Model Adapted to Evolving Policy," in ICML 2023

**Questions:**

1. I am very curious whether, in the latent space, there will indeed be unrealistic samples due to long rollouts that affect policy performance, just as in state-based MBRL methods. Based on this question, I also doubt whether the theoretical analysis from MBPO in Section 3.3 can be used to analyze DreamerV3.
2. Why does the rollout horizon in Figure 4 start long at the beginning of training and become shorter as training progresses? This is counter-intuitive. In the early stages of training, when the model is not very accurate, it should use short horizon rollouts. As the model learning progresses and becomes more accurate, the rollout horizon should gradually lengthen. I hope the paper can provide a reasonable explanation for this phenomenon.
3. Why does the performance curve for DreamerV3 only have 5 seeds, while the method proposed in this paper has 10 seeds? I believe this is not reasonable. Additionally, in Figures 4 and 7, the horizontal axis scales for DreamerV3 and the proposed method are not aligned, which can also influence the performance comparison.
4. Introducing the conservator will lead to additional training overhead. Can the paper provide a comparison of training time with Dreamer V3?

---

> ### Author Response · Authors · 2023-11-13
>
> Thank you for your helpful comments on our paper!
>
> * **Weakness1**:
>     * Thank you very much for the valuable revisions. As you said, our presentation is indeed not rigorous enough. We will revise the presentation and add relevant citations.
>     * Due to limited time and computational resources, we will conduct experiments on four games with linearly growing rollout length. We will add the experimental results in a follow-up reply.
> * **Weakness2**: We have taken note of these papers, but we believe that our approach and theirs have different emphases, despite the similarities in motivation. Our approach focuses on how to use the world model more safely for rollout by adjusting the rollout length, provided that model errors cannot be eliminated. Moreover, these two methods are based on a similar framework to MBPO, which requires storing imagined data and performing off-policy training, and cannot be directly migrated to dreamerv3, which performs real-time rollout and on-policy updating without saving imagined data. We briefly explain the reasons below.
>     * For M2AC, it needs to evaluate imagined rewards based on uncertainty. This requires the simultaneous training of multiple dynamical models, which is computationally prohibitive for Dreamer based on image inputs. And since the transition occurs in a latent space, ensemble's models need to be unified in one latent space, which is obviously difficult to do.
>     * For PDML,  its motivation is to learn more accurate dynamics models. It is essentially a data sampling method to achieve better model learning by modifying the sampling probability of different data. This is fundamentally different from our approach. Our approach considers how to utilize the model more safely and efficiently by adjusting the rollout length when the model error cannot be completely eliminated. Moreover, PDML needs to save its model parameters after each policy update, and hundreds of checkpoints need to be saved throughout the training process, which requires additional storage. And dreamerv3 is updated once for one step of interaction with the environment, which is clearly not applicable if we want to use PDML's method to estimate the policy shift, which requires saving tens of thousands of model parameters.
> * **Weakness3**: It is true that latent world model do not need to explicitly predict the next observation, but it is not protected from model errors. A key reason is that Dreamerv3 uses actor-critic architecture and needs to estimate the value of the latent state. Inaccurate latent state estimation can lead to problems with value estimation. And since the training of world model is based on limited real data, while the training of actor and critic are based entirely on imagined trajectories, this leads to the quality of imagined trajectories directly affecting the training of the actor and critic.
>
>     It is an oversight on our part not to add the relevant citation, which we will add. DMVE [1] shows the performance curves of Dreamer for different rollout length. When the rollout length is larger than 15, the performance shows a decreasing trend. And when the rollout length is 30, Dreamer almost completely loses its performance. We think this is enough to support our view.
>
> * **Weakness4**: Since the true sampled action distribution in continuous action space is difficult to estimate, we would like to focus on solving this problem in future research. However, this does not affect our aim to solve the problem in discretized action space. In fact there are many related algorithms focusing on discretized action space, such as IRIS [2] and TWM[3].
> * **Weakness5**: Since the y-axis is not scaled in Figure 7, some games with larger scores will appear to have limited improvement visually, and we show the percentage improvement in Figure 3 for a more intuitive comparison. Taking into account the variance of Atari, the data in Figure 3 are the average performance gains over 10 trainings, with each training performance test being the average result of 100 full episodes at the end of the training. This gives a more accurate reflection of the performance of the policy.
>
> Reference:
>
>       [1] Wang et al. "Dynamic Horizon Value Estimation for Model-based Reinforcement Learning". arXiv:2009.09593.
>       [2] Micheli et al. "Transformers are Sample Efficient World Models". ICLR2023
>       [3] Robine et al. "Transformer-based World Models Are Happy With 100k Interactions". ICLR2023.

---

> > ### Author Response · Authors · 2023-11-13
> >
> > * **Question1**:
> >     * We believe that there are undoubtedly long rollout samples in the latent space that can affect the performance of the policy. The example in Figure 5 is sufficient to illustrate this point. The world model mispredicts after t=10, leading to incorrect reward estimates, which affects the training of the policy. When interacting with the real environment, the agent may get stuck in its own fantasy and choose to move to the upper left in a state similar to the one at t=10 (actually down to get the reward).
> >     * There is indeed a lack of theoretical analysis of the Dreamer framework, but we believe that the theory of MBPO also has some applicability, although we do not provide a rigorous theoretical analysis. Dreamer uses the same branch rollout mechanism as mbpo. The big difference between the two is that MBPO is unrollingdirectly in the state space while Dreamer is unrolling in the latent space. But unlike TDMPC, Dreamer is based on explicit reconstruction, so its latent state can sufficiently reflect various details of the real state.
> > * **Question2**: We mention the way the conservator is trained in the experimental setup section in Section 4 "Due to the lack of data in the replay buffer and the instability of the encoder at the early stage, we train and apply the conservator after 10k steps to avoid overfitting". The average rollout length is large in the early stage because the output of the conservator is close to the average distribution in the beginning, resulting in the js-divergence with the actor that is almost always less than the threshold. The average rollout length does tend to decrease slightly during training. Our understanding is that since CRLA is judged with respect to the current policy, as the policy is updated, it may deviate from the historical action sampling distribution estimated by the conservator. The rollout length decreases gradually during this time. This is due to the fact that our approach is a conservative strategy that conservatively truncates the rollout as soon as there is a large change in policy with respect to the conservator. As the policy stabilizes, the sample distribution in the replay buffer also stabilizes, resulting in the output of the conservator stabilizing. At this time, the output of the conservator will be close to the output of the actor, which leads to a rise in the rollout length. However, since 100k interactions are not enough to make the policy converge completely, the average rollout length can only show a decreasing trend. It is worth noting that, according to the experimental observation, there are still a significant number of trajectories that reach the maximum rollout length in each rollout of the batch data. We will try to prove the above conjecture experimentally and reply later on.
> > * **Question3**:
> >     * The original data of Dreamerv3 comes from the author's [open-source code](https://github.com/danijar/dreamerv3) , which only provides data for 5 seeds. To ensure fairness, we respect the author's data. Considering the randomness of the Atari environment, we set 10 seeds to show the effectiveness of our algorithm with more certainty.
> >     * The horizontal axes in Figs. 4 and 7 do not correspond to each other, which is an oversight on our part, but the number of interaction steps is the same in both cases, only the meanings are different. In Fig. 7, it is the number of environment frames (frame skip=4), while in Fig. 4, it is the number of environment steps. We'll standardize on the number of environment frames as written in Dreamerv3.
> > * **Question4**:  This is a good proposal and we definitely should account for the extra time overhead compared to dreamerv3. We tested the time consuming to train a single environment on a single 3090. Our code is based on [Pytorch-Dreamerv3](https://github.com/NM512/dreamerv3-torch) for development. According to our tests, Dreamerv3 takes around 9.5 hours, while CLRA is around 11.5 hours, which is about 20% more time for training. The main computational overhead comes from the additional computation of the conservator network output at each rollout step.

---

### Official Review · Reviewer_stfm · 2023-10-31

**Soundness:** 2 fair
**Presentation:** 4 excellent
**Contribution:** 2 fair
**Rating:** 5
**Confidence:** 4

**Summary:**

This paper proposes a method to adaptively choose rollout length for model-rollouts model-based reinforcement learning to avoid compounding errors. The key idea is to learn a “conservator” that approximates the probability of distribution of action taken by the historical policy and then use the Jenson-Shannon divergence between the action selection distribution and the conservator to cut off the rollout when there is a large discrepancy. Empirical results on Atari100K benchmarks demonstrate the effectiveness of the proposed approach.

**Strengths:**

The paper is very well-written. The proposed algorithm is very well-motivated and easy to follow. The empirical evaluation is comprehensive (at least for tasks with discrete action spaces), with both quantitative and qualitative results. I appreciate the authors for providing a specific example on Ms Pacman to demonstrate how CRLA works.

**Weaknesses:**

Because the proposed method only works for environments with discrete action space, it becomes a major limitation that it cannot be applied to continuous control tasks.
Additionally, the performance improvement of the proposed approach is actually not quite significant based on the learning curve provided in Figure 7.

**Questions:**

1. The “Theoretical Analysis” section should better be named as Intuition or Theoretical Intuition since no rigorous analysis is provided, and it only explains the intuition of why cutting off the rollout based on disagreement between conservator and the policy.
Besides, it is based on the results of MBPO, which learns a dynamics model directly on the observation space, not a latent dynamics model.
2. Why does the rollout length get smaller towards the end of training based on Figure 4? Intuitively, as the dynamics model gets more accurate, it should be able to conduct longer rollouts. But for CRLA, the rollout length gets smaller throughout the training progress.

---

> ### Author Response · Authors · 2023-11-13
>
> Thank you for your helpful comments on our paper!
>
> * **Weaknesses**:
>     * Since the true sampled action distribution in continuous action space is difficult to estimate, we would like to focus on this issue in future research. But that doesn't stop us from focusing on discrete action space tasks. There are some model-based methods that also focus on discrete action spaces, such as some Transformer-based methods like IRIS.
>     * Since the y-axis is not scaled in Figure 7, some games with larger scores will appear to have limited improvement visually, and we show the percentage improvement in Figure 3 for a more intuitive comparison.
>
> * **Question1**:
>     Thanks a lot for providing valuable revisions. It is true that there is no rigorous theoretical analysis in the paper because there is currently no theoretical analysis specifically for Dreamer. We believe that the analysis of MBPO is also applicable to Dreamer to some extent because Dreamer also uses branch rollout to obtain imaginary trajectories, although Dreamer's training is on-policy while MBPO is off-policy. And although MBPO learns the dynamical model directly in the observation space, the analytical approach is also applicable to Dreamer because Dreamer reconstructs the observations and not just the rewards as TDMPC does. We hope to improve the theoretical analysis in our future work, and we also hope that more scholars will pay attention to the theoretical analysis of model-based methods in latent space.
>
> * **Question2**:
>     Since a cold start is detrimental to the training of the conservator, which may lead it to overfitting early on, we start training the conservator after 10k steps (40k frames) of interaction with the environment. Since the conservator's output is close to the average distribution at the beginning, which results in the computed js divergence being almost always less than the threshold. So the rollout length will be large, and as the conservator is trained, it will gradually be able to predict the historical action sampling distribution and adjust the rollout length. Because CRLA is judged with respect to the current policy, as the policy is updated, it may deviate from the historical action sampling distribution estimated by the conservator.  The rollout length decreases gradually during this time. This is due to the fact that our approach is a conservative strategy that conservatively truncates the rollout as soon as there is a large change in policy with respect to the conservator. As the policy stabilizes, the sample distribution in the replay buffer also stabilizes, resulting in the output of the conservator stabilizing. At this time, the output of the conservator will be close to the output of the actor, which leads to a rise in the rollout length. However, since 100k interactions are not enough to make the policy converge completely, the average rollout length can only show a decreasing trend. It is worth noting that, according to the experimental observation,  there are still a significant number of trajectories that reach the maximum rollout length in each rollout of the batch data. We will try to prove the above conjecture experimentally and reply later on.

---

> ### Comment · Reviewer_stfm · 2023-11-16
> **Response to Rebuttal**
>
> Thanks for the response! I have carefully read the response as well as other reviewers' comments. However, I don't find the response by the authors convincing at all. For the performance comparison, the authors claim that "some games with larger scores will appear to have limited improvement visually ." Could you then plot the learning curve (and perhaps an aggregated plot across all games) with respect to the Atari Human-normalized score? For the theoretical analysis, if the authors claim that they are providing theoretical justifications for their algorithms, I suggest the author rigorously present a version of their algorithms under the MBPO framework and carefully analyze the performance bounds. Otherwise, the version now only explains the intuition. Therefore, I would like to maintain my score.

---

> > ### Author Response · Authors · 2023-11-16
> >
> > Thank you sincerely for taking the time to read and respond to our paper and for your valuable comments. We will incorporate your feedback as we continue to refine this work in our future efforts.

---

### Official Review · Reviewer_pvE2 · 2023-11-03

**Soundness:** 2 fair
**Presentation:** 2 fair
**Contribution:** 2 fair
**Rating:** 5
**Confidence:** 3

**Summary:**

This paper introduces a novel approach for learning adaptive rollout lengths in model-based reinforcement learning, called "Conservative Rollout Length Adaptation" (CRLA). CRLA adjusts the length of model rollouts by truncating them if, at any given step, the policy significantly deviates from the action distribution in the training data, measured by Jensen–Shannon divergence. The method trains neural network, named “*conservator”*, on real transition data within the replay buffer to predict the action distribution. The truncation is triggered when the aforementioned divergence surpasses a predetermined threshold, at any step during the model rollout. Upon application to DreamerV3, CRLA has demonstrated performance enhancements on the Atari 100k benchmark.

**Strengths:**

- Outperforms the baseline on Atari 100k benchmark
- Adaptive model rollout horizon is relevant to a wide variety of MBRL methods and the proposed approach is straightforward

**Weaknesses:**

1. The evaluation is insufficient to assess the efficacy of the method: the approach is tested exclusively on the Atari 100k benchmark. It demonstrates a performance increase in 16 out of 28 games while underperforming in the remaining 12. Also, the threshold alpha is considered “crucial”, which is set heuristically. It remains to see whether such an approach is robust across varied environments.

2. Limited applicability: as pointed out in the paper, the approach is restricted to discrete action spaces, which narrows its applicability, especially when compared to the baseline DreamerV3 that accommodates both discrete and continuous actions.

3. Maybe I’m missing something, but I do not see why “Our method is computationally simpler compared to previous methods”.  Despite being conceptually straightforward, the method requires running a complete rollout up to the maximum length before any truncation occurs. Could you elaborate on the computational saving?

4. The presentation could be enhanced for better readability: the detailed exposition of the theorem from [Janner et. al 2019] seems to add limited value. One can simply mention the core components of the bound on the return of branched imagined rollouts and how they connect to the proposed method.

5. The rollout length is constrained inside an interval [5,16]. I wonder how would a random baseline for setting rollout length perform. Please also see the Question section below.

Minor:

- it is better to add supporting reference for this claim: “Previous works have found that even small model errors can be compounded by multi-step rollout and deviate the predicted state from the region where the model has high accuracy.” in Section 5.
- Typo in Sec 4.3: “, Our method” → “, our method”.

**Questions:**

1. How to set the interval of rollout length in practice?

2. I am curious about the performance of a simple random baseline where the rollout length is set randomly (say, uniformly), from the given range. Could the authors add it?

---

> ### Author Response · Authors · 2023-11-13
>
> Thank you for your helpful comments on our paper!
>
>
> * **Weakness1**:
>     * Since we focus on discrete action spaces, Atari100k is currently one of the most popular benchmarks, and several model-based methods have focused on Atari100k to test the effectiveness of the algorithm, e.g., IRIS[1], TWM[2]. Since Atari100k contains 26 games, its a difficult enough challenge to demonstrate the effectiveness of the algorithm.
>
>     * Although our algorithm still experienced performance degradation on 12 games, this is due to the fact that we used an automated threshold setting methodology which made it only necessary to set the same threshold parameter beta for all 26 environments. Since our algorithm does not essentially change the policy update rule, but only adjusts the rollout length, we could adjust the thresholds individually for those environments that show performance degradation, but this does not seem to be elegant enough.
>     * Our algorithm uses the same parameter \\(\\beta\\) to achieve a boost in 16 out of 26 Atari environments, which we believe reflects the robustness of our algorithm.
>
> * **Weakness2**:
>     The difficulty in estimating the true action sampling distribution in continuous action space makes it difficult to use our algorithm for continuous action space. However, this does not prevent us from focusing mainly on the discrete action space in this work. In fact some model-based approaches also focus on the discrete action space, such as IRIS. We leave improvements on continuous action spaces for future work.
>
> * **Weakness3**:
>     We apologize for the lack of rigor in our description and we will revise the description. We would like to emphasize the simplicity of the way of computation rather than the computational saving. Since our algorithm computes the respective rollout lengths for each rollout trajectory, to better utilize the GPU for batch inference, we chose to rollout the maximal lengths directly instead of removing the truncated trajectories. Due to the smaller number of parameters in the conservator network, the algorithm only takes about 20% more time compared to dreamerv3 (9.5h for dreamerv3 and 11.5h for CRLA), with about 3% more GPU memory usage.
>
> * **Weakness4**: Thank you very much for the proposed changes, we will make further revisions to this section.
>
> * **Questions1**: For the interval of rollout length, we tried a larger range of rollout but observed some performance degradation. This may require setting different thresholds for different environments, but we think this is not robust. Since it is difficult to accurately perform a longer rollout for environments based on image inputs, we empirically set the rollout length to [5,16]. This better emphasizes the balance of data quality and sample efficiency of our algorithm compared to a fixed step size of 15. We add a reference which illustrates the performance degradation of Dreamer after unfolding steps larger than 15 [3].
>
> * **Questions2**: This is a great suggestion and thank you very much for bringing it up. Due to limited computing resources, we will be testing this on four games, and the results of the experiment will be responded to later.
>
>
> \
> Reference:
>
>       [1]Micheli et al. "Transformers are Sample Efficient World Models". ICLR2023
>       [2]Robine et al. "Transformer-based World Models Are Happy With 100k Interactions". ICLR2023.
>       [3]Wang el al. "Dynamic Horizon Value Estimation for Model-based Reinforcement Learning".

---

> > ### Comment · Reviewer_pvE2 · 2023-11-20
> >
> > Thank you to the authors for their detailed response.
> >
> > I appreciate the reasonable answers provided to most of my initial concerns. However, I continue to have reservations regarding the absence of support for continuous action.
> >
> > Additionally, as highlighted by reviewers stfm and gVqn, the marginal performance improvement over DreamerV3, as shown in the learning curves in Fig. 7, raises another point of concern. Although I initially overlooked this aspect, I now find it to be a considerable drawback.
> >
> > The proposal to include further experimental results on alternative strategies for setting rollout lengths, such as a random schedule and the linearly growing schedule suggested by reviewer gVqn, appears promising.
> >
> > My score stands until I review the updated results (if any).

---

### Meta-Review · Area_Chair_pzLo · 2023-12-06

**Metareview:**

The authors propose a heuristic for dynamically adjusting the imagination rollout length of DreamerV3 over the course of training. The empirical benefits of the presented method (Fig 7) are believable but minor and it is unclear whether the critical hyperparameter of β=0.78 would be robust to new tasks. Conceptually, it is somewhat unclear why the rollouts should be stopped based on the actor distribution rather than the sampled action. Instead of truncating rollouts discretely, introducing a new trace coefficient for the lambda-return might be promising (see Retrace and related works). Other approaches to estimate when the world model goes OOD are left unexplored but should be compared to, especially ensemble disagreement (see Plan2Explore, LOMPO, etc). Exploring the design space more thoroughly and evaluating on additional environments other than Atari would significantly improve this paper.

**Justification For Why Not Higher Score:**

Empirical gains are small, evaluation only on Atari, lacking important baselines

**Justification For Why Not Lower Score:**

N/A

---

### Decision · Program_Chairs · 2024-01-16

Reject